

# Conversion of alpine pastureland to artificial grassland altered $CO_2$ and $N_2O$ emissions by decreasing C and N in different soil aggregates

Mei Zhang[1], Dianpeng Li[1], Xuyang Wang[1], Maidinuer Abulaiz[1], Pujia Yu[2], Jun Li[3], Xinping Zhu[1,4] and Hongtao Jia[1,4]

[1] College of Grassland and Environment Sciences, Xinjiang Agricultural University, Urumqi, China
[2] School of Geographical Sciences, Southwest University, Chongqing, China
[3] Akesu National Station of Observation and Research for Oasis Agro-ecosystem, Akesu, China
[4] Xinjiang Key Laboratory of Soil and Plant Ecological Processes, Urumqi, China

Corresponding author
Hongtao Jia, jht@xjau.edu.cn

## ABSTRACT

**Background**. The impacts of land use on greenhouse gases (GHGs) emissions have been extensively studied. However, the underlying mechanisms on how soil aggregate structure, soil organic carbon (SOC) and total N (TN) distributions in different soil aggregate sizes influencing carbon dioxide ($CO_2$), and nitrous oxide ($N_2O$) emissions from alpine grassland ecosystems remain largely unexplored.

**Methods**. A microcosm experiment was conducted to investigate the effect of land use change on $CO_2$ and $N_2O$ emissions from different soil aggregate fractions. Soil samples were collected from three land use types, i.e., non-grazing natural grassland (CK), grazing grassland (GG), and artificial grassland (GC) in the Bayinbuluk alpine pastureland. Soil aggregate fractionation was performed using a wet-sieving method. The variations of soil aggregate structure, SOC, and TN in different soil aggregates were measured. The fluxes of $CO_2$ and $N_2O$ were measured by a gas chromatograph.

**Results**. Compared to CK and GG, GC treatment significantly decreased SOC (by 24.9–45.2%) and TN (by 20.6–41.6%) across all soil aggregate sizes, and altered their distributions among soil aggregate fractions. The cumulative emissions of $CO_2$ and $N_2O$ in soil aggregate fractions in the treatments of CK and GG were 39.5–76.1% and 92.7–96.7% higher than in the GC treatment, respectively. Moreover, cumulative $CO_2$ emissions from different soil aggregate sizes in the treatments of CK and GG followed the order of small macroaggregates (2–0.25 mm) > large macroaggregates (> 2 mm) > micro aggregates (0.25–0.053 mm) > clay +silt (< 0.053 mm), whereas it decreased with aggregate sizes decreasing in the GC treatment. Additionally, soil $CO_2$ emissions were positively correlated with SOC and TN contents. The highest cumulative $N_2O$ emission occurred in micro aggregates under the treatments of CK and GG, and $N_2O$ emissions among different aggregate sizes almost no significant difference under the GC treatment.

**Conclusions**. Conversion of natural grassland to artificial grassland changed the pattern of $CO_2$ emissions from different soil aggregate fractions by deteriorating soil aggregate structure and altering soil SOC and TN distributions. Our findings will be helpful to develop a pragmatic management strategy for mitigating GHGs emissions from alpine grassland.

## INTRODUCTION

The global surface temperature is expected to increase by 1.8~4.0 °C by the end of the 21st century due to the increase of GHGs emissions (*IPCC, 2007*). The increase of $CO_2$ plays a key role in global warming as it contributes to approximately 63% of the total global warming effect. $N_2O$ is another primary GHG that largely depletes the ozone layer (*Ravishankara, Daniel & Portmann, 2009*). It has been estimated that $N_2O$ has 298 times higher global warming potential (GWP) than $CO_2$ over a 100 year time period (*IPCC, 2007*). Therefore, mitigation strategies for limiting the release of $CO_2$ and $N_2O$ from diverse soils are attracting increasing attention. As natural ecosystems are often reclaimed to farmland to satisfy humans' needs, several studies have demonstrated that land use change altered soil physicochemical properties as well as soil organic matter (SOM) content and quality (*Teixeira et al., 2019*). Consequently, these disruptions affected GHGs emissions from the soil (*Smith & Conen, 2004*; *Mcdaniel et al., 2019*; *Shakoor et al., 2020*). On the other side, soil aggregate is considered to be the structural unit of soils. It acts as the storage pool for soil nutrients and a living habitat for microorganisms that play an important role in controlling soil GHGs emissions (*Six et al., 2004*). Therefore, understanding, to what extent, land use change affecting $CO_2$ and $N_2O$ emissions from different soil aggregate fractions is vitally important to mitigate GHGs emissions from grassland.

Commonly, soil aggregate structure and the distribution of different aggregate fractions are sensitive to land use change and anthropologic disturbances (*Zhu, Deng & Shangguan, 2018*; *Shen et al., 2021*). Also, land use directly affected soil nutrient transformation (*Wei et al., 2013*; *Liu et al., 2014*). Previous studies have demonstrated that when grassland was converted to cropland, soil macroaggregate was easily destroyed (*Liu et al., 2014*) and the mineralization of SOM stored in inter-macroaggregate increased (*Six et al., 2004*), leading to the loss of soil C and N (*Linsler et al., 2013*). Moreover, the abilities of maintaining and supplying nutrients are different among different sizes of soil aggregates (*Arai et al., 2013*; *AşKın & Kızılkaya, 2009*). Soil nutrients (i.e., SOC, TN, and total phosphorus) in the larger aggregates were found to be generally higher than those in the smaller aggregates (*Liu et al., 2010*). Nevertheless, information remains limited on how land use change affects soil aggregate structure as well as the distributions of SOC and TN in different soil aggregate fractions.

Several studies demonstrated that the emissions of $CO_2$ and $N_2O$ from different sizes of soil aggregates were highly different because the quality and quantity of organic C and pore characteristics differed among soil aggregate sizes (*Perret et al., 1999*; *Diba, Shimizu & Hatano, 2011*; *Blagodatsky & Smith, 2012*; *Mangalassery et al., 2013*). Moreover, SOM decomposition was closely related to $N_2O$ emission as SOM mineralization was accompanied by the mineralization of soil nitrogen, whereas microorganisms participating in denitrification consumed organic C as an energy supply (*Kimura, Melling & Goh,*

*2012*). Some studies showed that the rates of soil C and N mineralization were higher in macroaggregates than in micro aggregates (*Mangalassery et al., 2013*; *Tian et al., 2015*). Similarly, *Wu et al. (2012)* found that $CO_2$ production was highest in >2 mm aggregate size, while lowest in $0-0.63$ mm aggregate size in a semiarid grassland of Inner Mongolia. However, other studies addressed that $CO_2$ emission in larger soil aggregates was lower than (*Sey et al., 2008*; *Drury et al., 2004*) or the same as (*Razafimbelo et al., 2008*) in smaller aggregates. *Uchida et al. (2008)* noted that the highest $N_2O$ emissions occurred in the smallest aggregates. These reflected that published results about different soil aggregate sizes on $CO_2$ and $N_2O$ emissions were inconsistent, and controversial findings were often reported. Therefore, studies about the impacts of land use change on GHGs emissions from different soil aggregate sizes in alpine grassland ecosystems are still needed to be further explored.

Alpine grassland is one of the main grassland types in Eurasia. It is not only a major animal husbandry development area, but also plays an important role in regulating global GHGs balance (*Li et al., 2012*). Bayinbuluk alpine grassland belongs to typical temperate alpine grassland located in the arid region of northwestern of China, and this region covers ca $2.3 \times 10^4$ $km^2$ area. To meet the demands of local animal husbandry, natural grassland areas are often reclaimed and converted to artificial grassland to grow forage grass. However, whether and how this land use change will affect $CO_2$ and $N_2O$ emissions from different soil aggregates in alpine grassland remains unclear. Therefore, a microcosm experiment was carried out to explore 1) the effect of different land use types on soil aggregate structure as well as the distributions of SOC and TN in different soil aggregate sizes, and 2) the influences of land use types on $CO_2$ and $N_2O$ emission fluxes from different soil aggregate sizes. We hypothesized that natural grassland converted to artificial grassland could affect the emissions of $CO_2$ and $N_2O$ from different soil aggregate sizes by altering the C and N contents in different soil aggregate fractions. The outcomes gained from the current study will shed useful insight into fragile alpine grassland management practices to protect the soil structure, maintain grassland sustainable development and alleviate GHGs emissions from the alpine grassland ecosystem.

## MATERIALS & METHODS

### Site and sampling

This study was conducted at the Bayinbuluk Grassland Ecosystem Research Station of the Chinese Academy of Science (42°53.1′N, 83°42.5′E, elevation 2,500 m a.s.l.). Bayinbuluk alpine grassland locates in the southern Tianshan Mountains, Xinjiang Uygur Autonomous Region in Central Asia. The mean annual temperature is −4.8 °C, and the mean annual precipitation is 265.7 mm with approximately 78.1% occurring from May to August. Plant species are dominated by *Stipa purpurea*, *Festuca ovina*, and *Agropyron cristatum* (*Li et al., 2012*). The soil type is a silty clay loam, which is defined as Mat-Cryic *Cambisols* based on the Chinese Soil Taxonomic System.

Three land use types of non-grazing natural grassland (CK), grazing grassland (GG), and non-grazing natural grassland converted to artificial grassland (GC) were established

in this study, and they are adjacent with each other. The CK has been fenced to exclusion grazing since 1984 (0.25 ha), the GG has a grazed density of two sheep ha$^{-1}$ year$^{-1}$ from October to April (100 ha), and the GC was reclaimed from non-grazing natural grassland to artificial grassland in 2014 (55 ha). CK and GG are dominated by *S. purpurea* and *F. kryloviana,* with vegetation coverage rates of 90% and 60%, respectively. The GC was consecutively cultivated with oat (*Avena sativa* L.) since 2014, and the above-ground biomass was annually harvested in the fall. Sheep manure (about $1.05 \times 10^5$ kg–$1.20 \times 10^5$ kg ha$^{-1}$) was applied at sowing in the spring once every three years, which was thoroughly incorporated in the soil surface layer (0–20 cm) with a rotary. No chemical fertilizers were used for more than the 5-year oat cultivation history. The annual irrigation column was approximately 400 to 600 mm through a drip tube in drip-irrigated condition.

For each land use type, four sampling sites were randomly chosen and treated as four replications. Soil samples were randomly collected from the 0–10 cm depth of topsoil with a soil auger (5.4 cm in diameter) in August 2018. The collected samples were sealed in plastic bags promptly, then carefully transported to the laboratory to acquire measurements of soil aggregate fractions and soil physicochemical properties.

## Soil aggregate fractionations and soil property analysis

Soil aggregate fractionation was performed using a wet-sieving method as described by *Feng et al. (2018)*. Briefly, a 100-g soil sample was passed through a series of three sieves (>2 mm, 2−0.25 mm, 0.25−0.053 mm, and <0.053 mm) by manually fractionating. After wet sieving, all soil aggregate fractions were collected and freeze-dried, then weighed and stored for soil chemical analysis. The <0.053 mm aggregate size was calculated by the residual value, namely, the total mass of the soil minus the mass of the other three aggregate fractions. The soil aggregate samples that were separated for the incubation experiment were dried at 40 ° C. Soil aggregate stability was expressed by the mean weight diameter (MWD) and geometric mean diameter (GMD), and calculated according to the method of *Nath & Lal (2017)*. Soil pH was measured in a 1:2.5 soil/water slurry by a digital pH meter (MP551, China). SOC content was determined by a potassium dichromate oxidation method (*Bao, 2000*), and the total N content was measured using an elemental analyzer (Vario EL III, Elementar, Hanau, Germany).

## Incubation experiments

Soil samples obtained from four aggregate fractions from different land use types (i.e., CK, GG, and GC) were used in incubation experiments. Each soil sample (treatment) repeated four times. An aliquot of a 30-g soil sample was placed in a glass jar (250 ml) and covered with porous film. All glass jars were incubated for 56 d at aerobic conditions (25 °C). Soil moisture was maintained 70% of water holding capacity (WHC) by adding distilled water every two to three days. Prior to soil incubation, all samples were pre-incubated for 3 d (25 °C) to activate microbial activity. Moreover, there were three blanks filled with quartz sand instead of soil samples.

The gaseous samples were collected at soil incubation times of 1, 3, 5, 7, 9, 16, 26, 36, 46, and 56 d, respectively. At each sampling time, all jars were sealed by a butyl rubber
septa for 24 h before sampling, subsequently, a syringe (100 ml) was employed to collect approximately 50 ml of gaseous sample from each jar. Then, the concentrations of $CO_2$ and $N_2O$ were immediately measured by a gas chromatograph (Agilent 7890A; Santa Clara, CA, USA). The concentrations of $CO_2$ and $N_2O$ were determined by an electron capture detector (ECD) and a flame ionization detector (FID), respectively. The detector temperatures were 250 °C for FID and 330 °C for ECD. The column temperature was maintained at 55 °C. The carrier gas flow rate was 25 ml min$^{-1}$ $N_2$ (Beijing AP BAIF Gases Industry Co., Ltd, China). After each sampling cycle was completed, the butyl rubber septa was removed to let the jars flush with ambient fresh air, and then re-sealed by a porous film again for the next measurement.

The fluxes of $CO_2$ and $N_2O$ emission were calculated using the following formula (*Ding, Sun & Huang, 2019*).

$$F = V \times (C_a - C_b) \times [273/(273 + T)] \times (1/22.4) \times M/(W \times t) \tag{1}$$

where F is the emission flux of $CO_2$ (ng $CO_2$–C g$^{-1}$ soil d$^{-1}$) or $N_2O$ (ng $N_2O$–N g$^{-1}$ soil d$^{-1}$), V is the jar headspace volume (L), $C_a$ and $C_b$ are the $CO_2$ (ppb) or $N_2O$ (ppb) concentrations in the soil sample and blank jars, respectively, T is the incubation temperature (°C), M is the molecular weight of $CO_2$–C (12) or $N_2O$-N (28), W is the soil dry weight (g), and t is the sealing time of the jars (d).

Cumulative flux (CF) of $CO_2$ and $N_2O$ during the whole incubation was computed as follows:

$$CF = \Sigma \left[ (F_{i+1} + F_i)/2 \right] / \times (D_{i+1} - D_i) \tag{2}$$

where F is the emission flux of $CO_2$ (mg kg$^{-1}$ soil d$^{-1}$) or $N_2O$ (ug kg$^{-1}$ soil d$^{-1}$), i is the i-th measurement, and $(D_{i+1} - D_i)$ is the number of days between two consecutive measurements.

### Statistical analysis

The statistical analysis was performed with SPSS 21.0 Statistics (IBM Software, Chicago, IL, USA). The SPSS procedure (one-way and two-way analysis of variance (ANOVA)) was used to analyze variance and to determine the statistical significance of the treatment effects. Duncan's multiple range test was used to compare means for each variable ($P < 0.05$). A simple linear regression model was used to examine correlations between $CO_2$ and $N_2O$ emissions from different soil aggregate class sizes.

## RESULTS

### Soil properties, soil aggregate fractions distribution and aggregate stability

Contents of SOC and TN in the GC treatment were significantly lower than in the treatments of CK and GG (Table 1). In the treatments of CK and GG, the dominant aggregate size was >2 mm, followed by 2−0.25 mm, 0.25−0.053 mm, and <0.053 mm (Fig. 1). Compared to the treatments of CK and GG, the proportion of >2 mm aggregate size in the GC treatment significantly decreased by 75.3% and 77.1%, respectively, whereas the

**Table 1 Soil properties, MWD and GMD under different land use types.**

| Land use | pH | SOC (g kg$^{-1}$) | TN (g kg$^{-1}$) | MWD | GMD |
|---|---|---|---|---|---|
| CK | 8.16 ± 0.07b | 49.84 ± 1.85a | 4.94 ± 0.22a | 1.35 ± 0.06a | 0.91 ± 0.09a |
| GG | 8.14 ± 0.03b | 51.63 ± 2.74a | 5.02 ± 0.29a | 1.30 ± 0.09a | 0.86 ± 0.16a |
| GC | 8.30 ± 0.07a | 32.65 ± 4.28b | 3.03 ± 0.33b | 0.76 ± 0.05b | 0.41 ± 0.04b |

**Notes.**

Data are mean ± standard deviation ($n = 4$).

Lowercase letters indicate differences in different land use types ($P < 0.05$).

CK, non-grazing natural grassland; GG, grazing grassland; GC, non-grazing natural grassland converted to artificial grassland; SOC, soil organic carbon; TN, total nitrogen; MWD, mean weight diameter; GMD, geometric mean diameter.

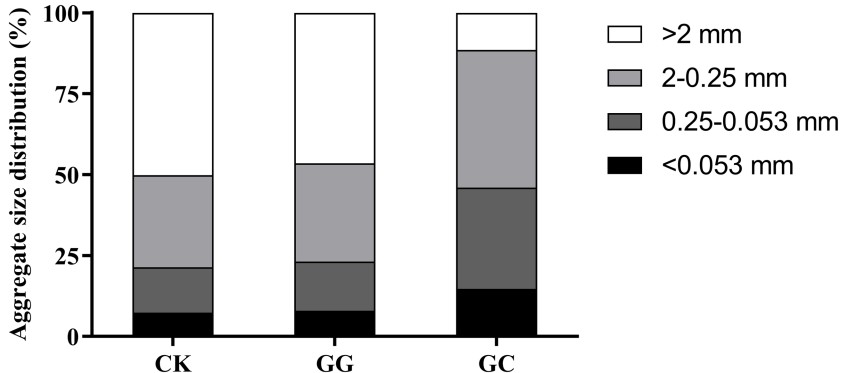

**Figure 1 The aggregate size distribution in different land use types.** CK, non-grazing natural grassland; GG, grazing grassland; GC, non-grazing natural grassland converted to artificial grassland.

proportion of 0.25−0.053 mm aggregate size increased by 106.6% and 122.9%. Moreover, the MWD and GMD values in the GC treatment were significantly lower than those in the treatments of CK and GG ($P < 0.05$) (Table 1). However, in terms of the variations of soil aggregate size distribution, MWD, and GMD, there were no significant differences between the treatments of CK and GG. These results indicate that non-grazing natural grassland converted to artificial grassland significantly deteriorated soil aggregate structure and reshaped the distribution of different aggregate fractions.

## Distributions of SOC and TN in different soil aggregate fractions

A significant lower SOC was achieved in the GC treatment across all aggregate fractions (Fig. 2). Compared to the treatments of CK and GG, the value of SOC in the GC treatment decreased by 26.20~45.2% and 24.9~41.2%, respectively. The highest SOC content consistently occurred in the aggregate size of 2−0.25 mm in the CK and GG treatments, followed by >2 mm. In contrast, the order of SOC content distribution in different aggregate sizes in the treatment of GC was >2 mm >2−0.25 mm >0.25−0.053 mm ><0.053 mm. The lowest SOC content was happened in <0.053 mm aggregate size across all aggregate fractions irrespective of land use types (Fig. 2). Likewise, a similar distribution pattern was found in soil TN across land use types. In a nutshell, the conversion of natural grassland to artificial grassland notably reduced soil SOC and TN and altered its distribution in different soil aggregate fractions as well.

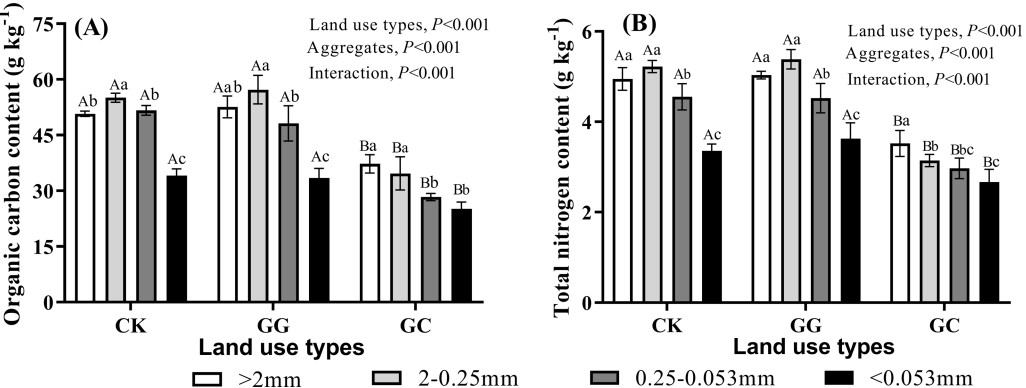

**Figure 2 Organic carbon (A) and total nitrogen contents (B) in soil aggregate fractions.** CK, non-grazing natural grassland; GG, grazing grassland; GC, non-grazing natural grassland converted to artificial grassland. Error bars represent standard deviation ($n = 4$). Capital letters indicate differences among land use types in the same aggregate size. Lowercase letters indicate differences in aggregate size fractions in the same land use type.

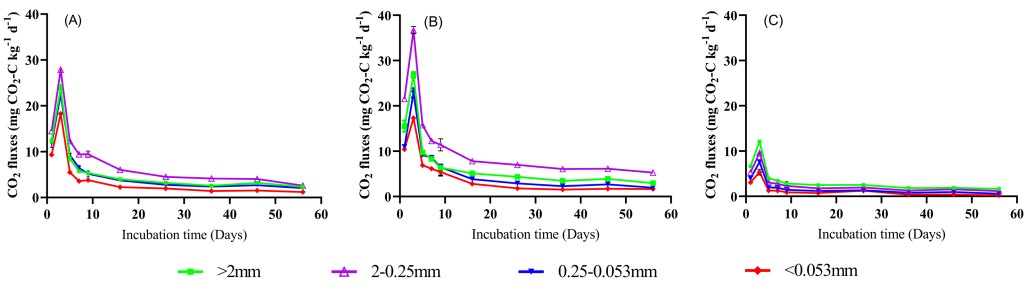

**Figure 3 The $CO_2$ emission flux in soil aggregate fractions during a 56-day incubation.** (A) CK, non-grazing natural grassland. (B) GG, grazing grassland. (C) GC, non-grazing natural grassland converted to artificial grassland. Error bars represent standard deviation ($n = 4$).

## $CO_2$ emission fluxes and cumulative emissions

Despite land use pattern, soil $CO_2$ emission fluxes initially increased sharply from 1 d to 3 d, then steadily decreased during the rest time of incubation (3–56 d) across all soil aggregate sizes (Fig. 3). The influence of land use change on $CO_2$ flux across all aggregate fractions followed the sequence of GG >CK >GC. For example, the averaged $CO_2$ fluxes (across all aggregate fractions) in the GG treatment was three times higher than that in the treatment of GC. During the whole incubation period, the highest $CO_2$ fluxes in the treatments of CK and GG were happened in 2−0.25 mm aggregate size across all aggregates fractions, followed by >2 mm aggregate size, while the lowest $CO_2$ emission fluxes occurred in <0.053 mm aggregate size. Compared to the CK treatment, the averaged $CO_2$ fluxes in 2−0.25 mm and >2 mm aggregate sizes increased by 37.0 and 21.8%, respectively, in the treatment of GG ($P < 0.05$, Figs. 3A and 3B). In contrast, $CO_2$ emission fluxes in the treatment of GC decreased as the aggregate size decreasing ($P < 0.05$, Figs. 3A and 3B).

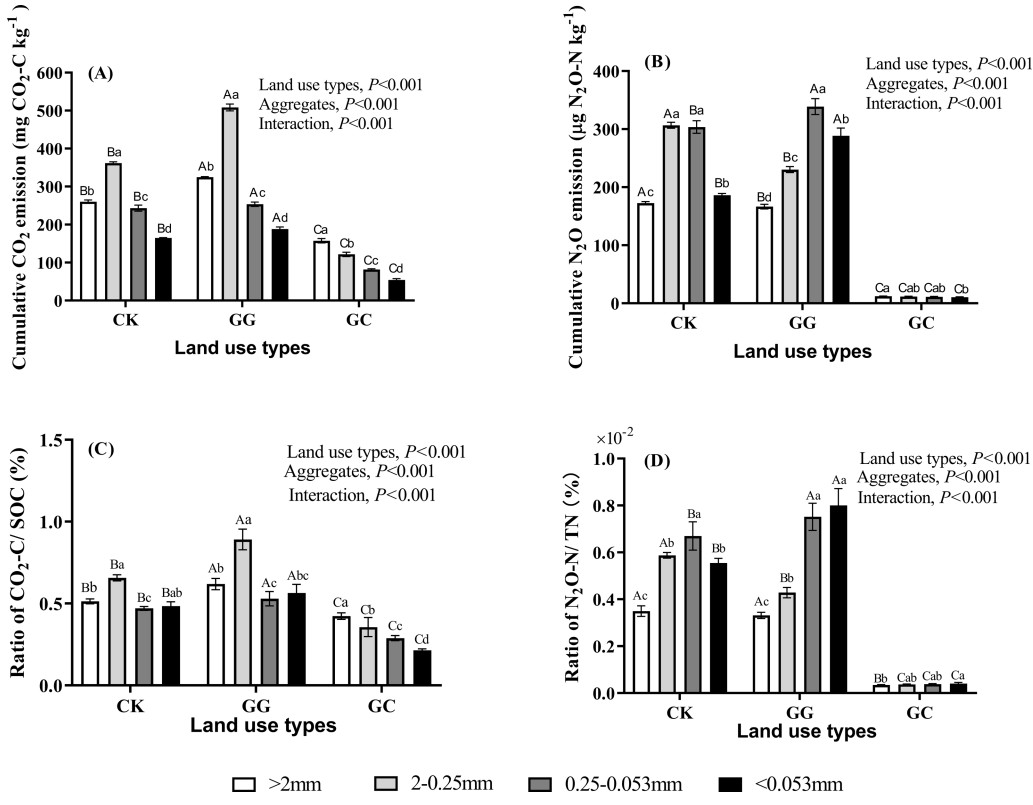

**Figure 4** **The cumulative emission of $CO_2$ (A) and $N_2O$ (B) in soil aggregate fractions under different land use types at the 56 day. Ratios of $CO_2$-C/SOC (C) and $N_2O$-N/TN (D) in soil aggregate fractions under different land use.** SOC, soil organic carbon; TN, total nitrogen. CK, non-grazing natural grassland; GG, grazing grassland; GC, non-grazing natural grassland converted to artificial grassland. Error bars represent standard deviation ($n = 4$). Capital letters indicate differences among land use types in the same aggregate size. Lowercase letters indicate differences in aggregate size fractions in the same land use type.

Moreover, land use type, soil aggregate size, and interactions significantly affected cumulative $CO_2$ emission (Fig. 4A). Cumulative $CO_2$ emission gradually increased as time prolonging, and then kept a stable trend at later stages of incubation (Fig. S1). The cumulative $CO_2$ emissions in three land use types (56 d) were 164.9–361.7 mg kg$^{-1}$ (CK), 188.1–508.3 mg kg$^{-1}$ (GG), and 54.0–157.4 mg kg$^{-1}$ (GC), respectively. Notably, in the treatment of GG, the highest cumulative $CO_2$ emission was found in 2−0.25 mm aggregate size among four aggregate sizes, and it increased by 40.5% and 318.3%, respectively, compared to the treatments of CK and GC. Additionally, the average ratios of $CO_2$ emission account for SOC in all aggregate fractions followed the order of GG >CK >GC (Fig. 4C). In the treatments of CK and GG, the highest ratio of $CO_2$ emission account for SOC was observed in 2−0.25 mm aggregate size. In contrast, in the treatment of GC, the ratios of $CO_2$ emission account for SOC decreased as the aggregate size decreasing.

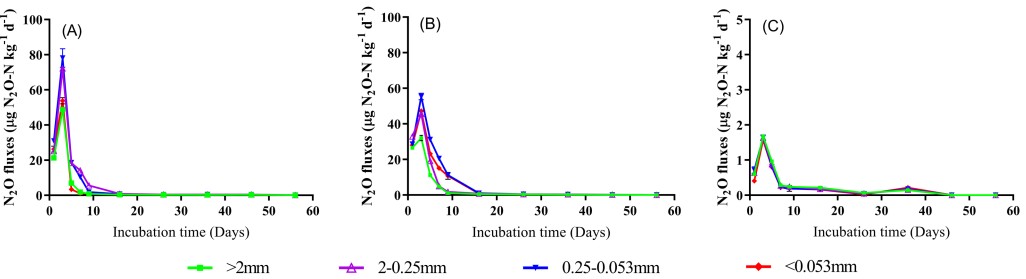

**Figure 5 The N₂O emission flux in soil aggregate fractions during a 56-day incubation.** (A) CK, non-grazing natural grassland. (B) GG, grazing grassland. (C) GC, non-grazing natural grassland converted to artificial grassland. Error bars represent standard deviation ($n = 4$).

## N₂O emission fluxes and cumulative emissions

As shown in Fig. 5, N₂O emission fluxes significantly increased from 1 d to 3 d, and then showed a decreasing trend during the rest time of incubation (3–56 d) across all soil aggregate sizes despite land use types. Compared to the treatments of CK and GG, N₂O emission flux was consistently and significantly lower under the GC treatment across all soil aggregate sizes during whole incubation time ($P < 0.05$, Fig. 5). The averaged N₂O fluxes (across incubation time) in the CK followed order of 0.25−0.053 mm >2−0.25 mm ><0.053 mm >>2 mm, and the order of the averaged N₂O fluxes in the treatment of GG was 0.25−0.053 mm ><0.053 mm >2−0.25 mm >>2 mm. However, N₂O fluxes in the treatment of GC decreased as the aggregate size decreasing. Furthermore, cumulative N₂O emission gradually increased as time prolonging, and then showed a constant trend at later stages of incubation (Fig. S2). The tendency of the variation of cumulative N₂O emission at 56 d was similar to N₂O emission fluxes (Fig. 4B). The percentage of soil TN lost through N₂O emissions in the treatments of CK and GG was significantly higher than in the treatment of GC across all aggregate fractions (Fig. 4D). Regardless of land use types, the percentage of soil TN lost through N₂O emissions was higher in >0.25 mm aggregate size than in <0.25 mm aggregate size.

## DISCUSSION

### The distributions of SOC and TN in different aggregate fractions

In our study, non-grazing natural grassland converted to artificial grassland significantly reduced SOC and TN contents across all soil aggregate fractions, which was in accordance with previous studies (*Li et al., 2007*; *Linsler et al., 2013*). This phenomenon mainly ascribed to that reclamation and artificial cultivation practices destroyed the structure and stability of soil aggregates (Fig. 1, Table 1) and accelerated the decomposition of SOC which initially protected by macroaggregate. What's more, the process of SOC degradation usually paralleled with soil N mineralization, thus lead to synchronous loss of soil C and N (*Conant et al., 2007*; *Häring et al., 2013*). Our results showed the highest contents of SOC and TN were observed in macroaggregates (>0.25 mm) across all land use types (Fig. 2), which were coincident with the principles of aggregate hierarchy (*Six et al., 2004*). It was

worth noting that the greatest values of SOC and TN in the non-grazing natural grassland and grazing grassland occurred in 2−0.25 mm aggregate size. In contrast, the highest SOC and TN appeared in >2 mm aggregate size in the artificial grassland (Fig. 2), indicating land use change altered SOC and TN distributions among different aggregate fractions in alpine grassland.

### The emissions of $CO_2$ and $N_2O$ in different soil aggregate fractions

Our findings showed that land use change significantly affected $CO_2$ and $N_2O$ emissions from different soil aggregate sizes (Figs. 3, 4 and 5). The averaged $CO_2$ or $N_2O$ emission rate (in 56 d) in all aggregate size fractions was significantly higher in non-grazing natural grassland and grazing grassland soils (CK and GG) than in artificial grassland soil (GC). The difference can probably due to the continuous supply of labile organic C (easily oxidizable organic C) because an abundant of grass debris and root residues were returned to soil from perennial vegetation under the natural grassland condition, but that was greatly reduced under the cultivation condition (artificial grassland) (Liu et al., 2014). Similarly, a meta-analysis showed that no-tillage management practice significantly increased $CO_2$, $N_2O$ and $CH_4$ emissions by 7.1%, 11.9% and 20.8 compared with conventional tillage, respectively (Shakoor et al., 2021). Another possible explanation may be attributed to that cultivation practices cause the mineralization of easily decomposable SOC and a higher proportion of chemically stable SOC in the artificial grassland soil compared to the natural grassland soil (Cambardella & Elliott, 1994; Six et al., 1998). Our data showed that the fluxes of $CO_2$ and $N_2O$ emissions reached the peak point at the 3rd day across all treatments, then both steadily decreased (Figs. 3 and 5). This could be explained by priming effect at the beginning of incubation. Commonly, a large amount of labile C and available N and soil microbial activity can be promptly increased in a short time when soil subjected to dry-rewetting cycle, as a result, soil biological reaction processes such as soil organic mineralization, nitrification and denitrification were stimulated, and thus the emissions of both $CO_2$ and $N_2O$ were increased (Borken & Matzner, 2009; Beare, Gregorich & St-Georges, 2009). Following this, published literature demonstrated that soil GHGs emission rate generally reached the maximum at the 1 to 7th day after incubation (Borken & Matzner, 2009; Diba, Shimizu & Hatano, 2011; Tao et al., 2021). Therefore, the decreases in $CO_2$ emission fluxes along with incubation time after 3rd day can be primarily attributed to the stock of soil available organic C is apt to be easily depleted with time (Usman, Kuzyakov & Stahr, 2004; El-naggar et al., 2015). In addition, the averaged $CO_2$ emission rate across all aggregate fractions in the treatment of GG was higher than in the CK treatment (Figs. 3A and 3B). A similar result was also observed by Wu et al. (2012) who found that heavy grazing and continuous grazing increased $CO_2$ emissions, microbial biomass carbon and dissolved organic carbon concentration across all aggregate fractions in comparison with non-grazing treatment. Thus, it can be assumed that natural grassland has more labile C than artificial grassland in the alpine grassland ecosystem, when natural grassland soil is reclaimed, this anthropogenic disturbance/practices will inevitably release a greater amount of GHGs (Fig. 6).

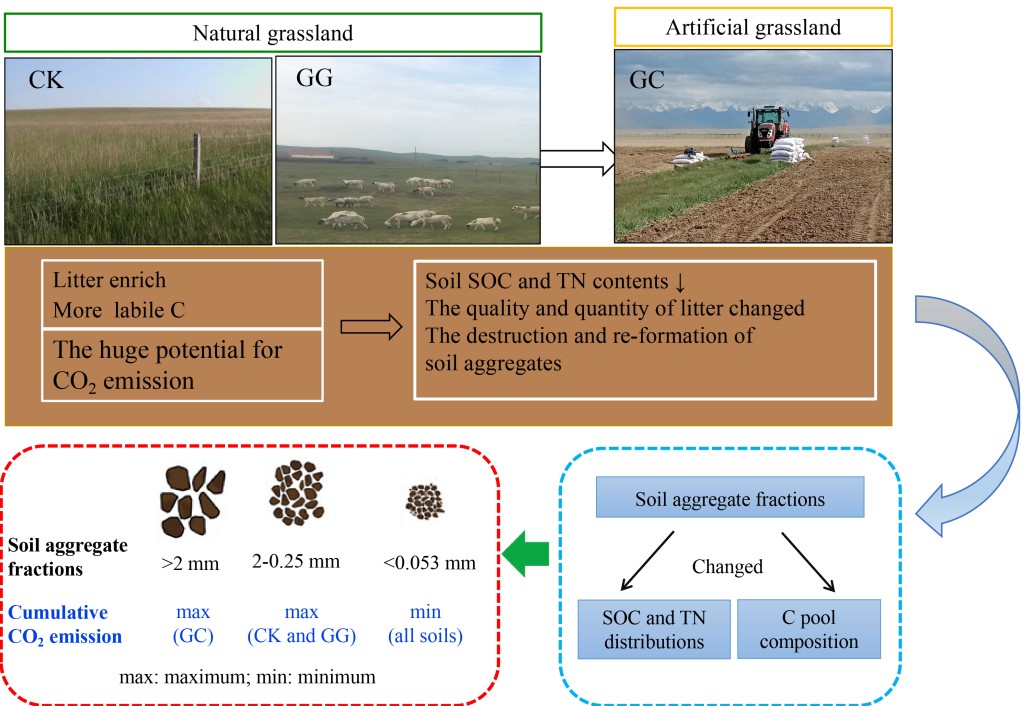

**Figure 6  Conceptional diagram in explaining the mechanisms of land use change on soil CO$_2$ emission among soil aggregate fractions.** CK, non-grazing natural grassland; GG, grazing grassland; GC, non-grazing natural grassland converted to artificial grassland.

The macroaggregate fractions (>0.25 mm) had a higher CO$_2$ emission rate than micro aggregate fractions (<0.25 mm) across all land use types (Fig. 3). This was likely due to a large number of free particulate organic carbon (POC) existed in the macroaggregates which may provide sufficient unprotected and labile C and N (*Drury et al., 2004*; *Sarker et al., 2018*). Our results showed that the lowest CO$_2$ emission rate was observed in <0.053 mm aggregate size (silt and clay fraction) across all land use types (Fig. 3). Because organic C was physically protected by silt and clay particles rather than coarse soil particles (*Chung et al., 2010*), with turnover times varied from 400 to 1,000 years (*Buyanovsky, Aslam & Wagner, 1994*).

Interestingly, we found that the greatest CO$_2$ emission rate and cumulative CO$_2$ emission were observed in 2−0.25 mm aggregate size in the non-grazing natural grassland and grazing grassland soils, whereas that occurred in >two mm aggregate size in the artificial grassland soil (Figs. 3A and 3B). This occurrence can probably be attributed to that the conversion of non-grazing natural grassland to artificial grassland resulted in reallocation of SOC and TN among different aggregate sizes through reformatting soil aggregates (Fig. 7) (*Puget, Chenu & Balesdent, 2000*). Therefore, the variations of CO$_2$ emissions from different sizes of aggregates in natural grassland were significantly differentiated from artificial grassland. Meanwhile, the higher SOC and TN contents occurred in 2−0.25 mm or >2 mm aggregate size (Fig. 2), these can provide energy to microbes, and promote SOM mineralization

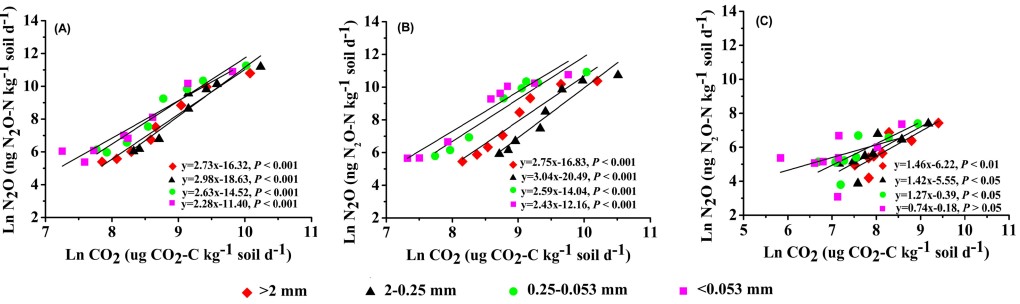

**Figure 7** Relationship between soil $N_2O$ and $CO_2$ emission in soil aggregate fractions under different land use types. (A) CK, non-grazing natural grassland. (B) GG, grazing grassland. (C) GC, non-grazing natural grassland converted to artificial grassland.

(*Saggar et al., 2013*; *Zhang et al., 2014*). In agreement with ours findings, *Chen et al. (2017)* addressed that the difference in $CO_2$ emissions from different aggregates was closely associated with the contents and distributions of soil organic C and N in soil aggregates. This was further evidenced by our correlation analysis results (Table 2), and supported our hypothesis that the difference in SOC and TN contents in different aggregate fractions can affect soil aggregate C mineralization under the conversion of natural grassland to artificial grassland condition. On the other hand, more sufficient plant debris and root residues in natural grassland may lead to the composition of SOC among soil aggregate sizes significantly distinguished from the treatment of GC (*Yamashita et al., 2006*; *Ding et al., 2014*). A recent study found that soil physicochemical factors (i.e., SOC, total porosity, and pH) in soil aggregate fractions affected bacterial and fungal communities, which further influenced soil biological respiration among different soil aggregate fractions (*Yang, Liu & Zhang, 2019*). In addition, in this study, soil pH was a significant difference between non-grazing natural grassland, grazing grassland, and artificial grassland (Table 1), which may be another reason for the explanation on why $CO_2$ emissions differ between non-grazing natural grassland and artificial grassland among the soil aggregate fractions. Nevertheless, how different forms of C pool distribution and microbial activity in different soil aggregate fractions response to land use conversion from non-grazing natural grassland to artificial grassland, and further influence GHGs emission from a fragile alpine grassland need to be deciphered.

Our results showed that the pattern of $NO_2$ emissions among four aggregate fractions was significant different between land use types. Many studies have found that the $N_2O$ emission rate differed among different soil aggregate sizes (*Bandyopadhyay & Lal, 2014*; *Plaza-Bonilla, Cantero-Martinez & Alvaro-Fuentes, 2014*), and which was closely correlated with $CO_2$ emissions (*Ding, Sun & Huang, 2019*). In agreement with these findings, our data showed that the $N_2O$ emission rate was positively associated with the $CO_2$ emission rate. Moreover, the regularity of $N_2O$ emissions in four aggregate fractions was consistent with $CO_2$ emissions from corresponding aggregate fractions in artificial grassland (GC treatment) (Figs. 5 and 7). Our findings were supported by *Robinson et al. (2014)* who reported that patterns of $N_2O$ emissions were affected by

**Table 2** Pearson's correlation coefficients between the cumulative emissions of $CO_2$ and $N_2O$ with soil chemical properties in soil aggregates across three land use types.

| Land use | Variable | SOC | TN | C/N ratio |
|---|---|---|---|---|
| CK | $CO_2$ emission | 0.860[**] | 0.870[**] | 0.148 |
| | $N_2O$ emission | 0.615[*] | 0.434 | 0.565[*] |
| GG | $CO_2$ emission | 0.815[**] | 0.849[**] | 0.481 |
| | $N_2O$ emission | −0.433 | −0.523[*] | −0.133 |
| GC | $CO_2$ emission | 0.881[**] | 0.816[**] | 0.451 |
| | $N_2O$ emission | 0.532 | 0.701[**] | 0.061 |
| All lands | $CO_2$ emission | 0.890[**] | 0.907[**] | 0.268 |
| | $N_2O$ emission | 0.649[**] | 0.671[**] | 0.153 |

**Notes.**

SOC, soil organic carbon; TN, total nitrogen; CK, non-grazing natural grassland; GG, grazing grassland; GC, non-grazing natural grassland converted to artificial grassland.

$n = 16$ for each land use type, respectively. $n = 48$ for correlations of all lands.

[*]$P < 0.05$.

[**]$P < 0.01$.

the aggregate size, and the higher $N_2O$ emissions were observed in the large and medium aggregates (1–2, 2–4 and 4−5.6 mm size). The possible explanation is that SOC mineralization consumes a great fraction of oxygen, then results in an anaerobic zone, promotes $N_2O$ production through denitrification (*Pare & Bedard-Haughn, 2013*). Normally, soil $N_2O$ is primarily produced by the denitrification process, which is influenced by soil microorganisms and physicochemical properties (*Diba, Shimizu & Hatano, 2011*; *Chapuis-Lardy et al., 2007*) while the higher SOC provides enough energy for the denitrifies. On the other hand, an anaerobic "hot spots" was easily created by larger aggregates within the inner of aggregate due to their possessing sufficient micropores, which can hold more water (*Sexstone et al., 1985*; *Khalil, Renault & Mary, 2005*; *Diba, Shimizu & Hatano, 2011*). In agreement with this, *Drury et al. (2004)* found that the production of $N_2O$ increased as aggregate size increasing. In this study, the $N_2O$ emission rate from macroaggregate in non-grazing natural grassland and grazing grassland soils was lower than that in micro aggregate and clay and silt, especially in the grazing grassland. *DeCatanzaro, Beauchamp & Drury (1987)* reported that the addition of alfalfa residue to soils facilitated dissimilatory nitrate reduction and reduced the $N_2O$ production (denitrification) under anaerobic conditions through a [15]N tracer study. This was likely due to before the produced $N_2O$ in macroaggregate was released from soil, it was consumed within soil and eventually transformed to $N_2$ via denitrification (*Chapuis-Lardy et al., 2007*; *Gu et al., 2013*). Another possible explanation is that the quantity of litter and root residue was greater in macroaggregates than in microaggregates, which strengthened dissimilatory nitrate reduction. Therefore, the $N_2O$ emissions from different aggregate fractions were affected by different edaphic factors, and the alterations of soil aggregate fractions affected by land use on $N_2O$ emission were still elusive and need to be further studied.

## CONCLUSIONS

In summary, grazing had no significant influence on the variations of soil aggregate structure, SOC, or TN contents and their distribution in different soil aggregate fractions. However, conversion from natural alpine grassland to artificial grassland notably deteriorated soil aggregate structure and significantly decreased soil SOC and TN across all soil aggregate sizes. Moreover, the conversion of natural grassland to artificial grassland altered the distributions of SOC and TN among different aggregate sizes, which resulted in the significant differences in $CO_2$ emissions from different soil aggregate fractions. The greatest $CO_2$ emission occurred in 2−0.25 mm aggregate size in natural grassland and grazing grassland, whereas that happened in >2 mm aggregate size in the artificial grassland. Besides, alpine grassland converted to artificial grassland influenced $N_2O$ emissions. Unfortunately, the $N_2O$ emissions from different soil aggregate fractions in either natural grassland, grazing grassland or artificial grassland were largely different from each other. Taken together, based on the aggregate scale, our findings provide a valuable insight into understanding the influences of land use change on soil aggregate structure, SOC and TN, and soil GHGs emissions in an alpine grassland ecosystem. Therefore, we suggest that the anthropogenic disturbance and conversion of natural grassland to artificial natural grassland (reclamation) should be prohibited to sequestrate soil organic carbon, maintain a better soil structure to improve soil quality, and mitigate GHGs emission from fragile alpine grassland ecosystems.

## ACKNOWLEDGEMENTS

We sincerely thank Professor Guixin Chu for helpful and valuable comments on the draft manuscript. We would like to thank Bayinbuluke Steppe Research Station, Xinjiang Institute of Ecology and Geography, Chinese Academy of Sciences for the sampling support. We acknowledge TopEdit LLC for the linguistic editing and proofreading during the preparation of this manuscript.

### Funding

This work was supported by the National Natural Science Foundation of China (No. 31560171 and 31960258). The funders had no role in study design, data collection and analysis, decision to publish, or preparation of the manuscript.

### Grant Disclosures

The following grant information was disclosed by the authors:
The National Natural Science Foundation of China: 31560171, 31960258.

### Competing Interests

The authors declare there are no competing interests.

## Author Contributions

- Mei Zhang conceived and designed the experiments, performed the experiments, analyzed the data, prepared figures and/or tables, authored or reviewed drafts of the paper, and approved the final draft.
- Dianpeng Li performed the experiments, analyzed the data, prepared figures and/or tables, authored or reviewed drafts of the paper, and approved the final draft.
- Xuyang Wang and Maidinuer Abulaiz performed the experiments, analyzed the data, prepared figures and/or tables, and approved the final draft.
- Pujia Yu, Jun Li and Xinping Zhu analyzed the data, authored or reviewed drafts of the paper, and approved the final draft.
- Hongtao Jia conceived and designed the experiments, analyzed the data, prepared figures and/or tables, authored or reviewed drafts of the paper, and approved the final draft.

## Data Availability

The raw measurements are available in the Supplementary File.

## Supplemental Information

Supplemental information for this article can be found online at http://dx.doi.org/10.7717/peerj.11807#supplemental-information.

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
