# Peer review of "Conversion of alpine pastureland to artificial grassland altered CO2 and N2O emissions by decreasing C and N in different soil aggregates"

_PeerJ, doi:10.7717/peerj.11807_

## Round 0.1 · original submission · Major Revisions

Reviewers have now commented on your manuscript and they have suggested that you revise your manuscript. I invite you to revise the manuscript by giving due consideration to the comments and suggestions from reviewers. I also ask you to carefully check for any grammatical and typographic mistakes.

·

Basic reporting

Findings of this study provide a valuable insight into understanding the influences of land use change on soil aggregate structure, SOC and TN, and soil GHGs emissions in an alpine grassland ecosystem.

Experimental design

Methodology is good and clearly presented.

Validity of the findings

Result section in the abstract should be improved. Author wrote basic and general results, however results whit actual data would be good. moreover, the abstract should be reduced.
overall results are presented very well.

Additional comments

The manuscript entitled “Deterioration of aggregate structure and decrease of soil organic C and N caused by alpine pastureland converted to artificial grassland altered CO2 and N2O emissions pattern” is interesting and suitable for publication but several mistakes in the manuscript should be corrected before acceptance.
Specific comments
• Authors should carefully recheck the manuscript for the typos and also write the meaning of all acronyms because there are some which are not presented.
• All the tables and figures are clearly presented but authors must check all the figure pixels again. Some of them are not very clear.
• I suggest that authors check the bibliographic references in the main text and revise the format of documents to meet the requirement of the Journal.
• In abstract, please add more information about methods of the experiment.
• In abstract, please don't write the same conclusion lines as they were written in the main conclusion section.
• In abstract, line 36-41; it’s a general result, author should present data and justify it.
• Introduction, Line 76: “use GHGs abbreviation”.
• Introduction, Line 83: correct the spell of different.
• Methodology Line 85: what is TP stands for???
• Discussion should also be improved critically. Don't write the results in the discussion section, discus them with strong logic.
• Please recheck whole manuscript very carefully, particularly English language and remove all the typo mistakes.
• Some of the papers that would be interesting to compare and cite in this current study are, for example.

Shakoor, A., Shakoor, S., Rehman, A., Ashraf, F., Abdullah, M., Shahzad, S. M., ... & Altaf, M. A. (2020). Effect of animal manure, crop type, climate zone, and soil attributes on greenhouse gas emissions from agricultural soils—A global meta-analysis. Journal of Cleaner Production, 124019.

Shakoor, Awais, Muhammad Shahbaz, Taimoor Hassan Farooq, Najam E. Sahar, Sher Muhammad Shahzad, Muhammad Mohsin Altaf, and Muhammad Ashraf. "A global meta-analysis of greenhouse gases emission and crop yield under no-tillage as compared to conventional tillage." Science of The Total Environment 750 (2021): 142299

Reviewer 2 ·

Basic reporting

In this paper the authors have made an attempt to study the effect of conversion of alpine pasture land to artificial grassland on distribution of soil aggregates and aggregate associated C and N and also emissions of CO2 and N2O from these aggregates. This is an interesting study. This paper may be accepted for publication with major revisions as suggested in Specific comments and also on the body of the text.

Experimental design

Results
Page 16 Line 233: Percentage of SOC lost CO2 emissions may be computed.

Page 16 Line 256: Percentage of soil TN lost through N2O emissions may be computed. The data on cumulative N2O fluxes may be presented.

Discussion
Page 20 Line 328: C/N ratio in different aggregate fractions may be computed and correlated with the CO2 and N2O emissions. Because C/N ratio indicates the relative lability of SOC to microbial decomposition
Page 20 Line 334-336: Estimated carbon input under different treatments may be computed to support this hypothesis.

Table 1: SOC and TN of bulk soil may be presented.
Table 2: The number of data points used in correlation may be indicated. Correlation between CO2 and N2O emission with C/N ratio may be attempted.

Figure 3: The peak value of CO2 emissions may be discussed.
Figure 4: The peak value of N2O emissions may be discussed.

Validity of the findings

It is an interesting study and the conclusions are well stated, linked to original research questions and limited to supporting results.

Additional comments

Please address the Specific comments and also the comments made on the body of the text.

Annotated reviews are not available for download in order to protect the identity of reviewers who chose to remain anonymous.

---

## Round 0.2 · accepted · Accept

After careful consideration of your revised manuscript and recommendations from our reviewer(s), I am pleased to inform you that your revised manuscript is accepted for publication.

·

Basic reporting

Clear

Experimental design

Clear

Validity of the findings

Good

Additional comments

Authors have answered all the concerns. Paper can be accepted for publication.